# Development of Good Manufacturing Practice-Compatible Isolation and Culture Methods for Human Olfactory Mucosa-Derived Mesenchymal Stromal Cells

**DOI:** 10.3390/ijms25020743

**Published:** 2024-01-06

**Authors:** Christopher J. Kelly, Susan L. Lindsay, Rebecca Sherrard Smith, Siew Keh, Kyle T. Cunningham, Katja Thümmler, Rick M. Maizels, John D. M. Campbell, Susan C. Barnett

**Affiliations:** 1School of Infection and Immunity, 120 University Place, Glasgow G12 8TA, UK; christopher.kelly@glasgow.ac.uk (C.J.K.); susan.lindsay@glasgow.ac.uk (S.L.L.); rick.maizels@glasgow.ac.uk (R.M.M.);; 2New Victoria Hospital, 55 Grange Road, Glasgow G42 9LF, UK; 3Tissues Cells and Advanced Therapeutics, SNBTS, Jack Copland Centre, Edinburgh EH14 4BE, UK

**Keywords:** mesenchymal stromal cells, myelination, good manufacturing practice, cellular therapy, olfactory mucosa-derived

## Abstract

Demyelination in the central nervous system (CNS) resulting from injury or disease can cause loss of nerve function and paralysis. Cell therapies intended to promote remyelination of axons are a promising avenue of treatment, with mesenchymal stromal cells (MSCs) a prominent candidate. We have previously demonstrated that MSCs derived from human olfactory mucosa (hOM-MSCs) promote myelination to a greater extent than bone marrow-derived MSCs (hBM-MSCs). However, hOM-MSCs were developed using methods and materials that were not good manufacturing practice (GMP)-compliant. Before considering these cells for clinical use, it is necessary to develop a method for their isolation and expansion that is readily adaptable to a GMP-compliant environment. We demonstrate here that hOM-MSCs can be derived without enzymatic tissue digestion or cell sorting and without culture antibiotics. They grow readily in GMP-compliant media and express typical MSC surface markers. They robustly produce CXCL12 (a key secretory factor in promoting myelination) and are pro-myelinating in in vitro rodent CNS cultures. GMP-compliant hOM-MSCs are comparable in this respect to those grown in non-GMP conditions. However, when assessed in an in vivo model of demyelinating disease (experimental autoimmune encephalitis, EAE), they do not significantly improve disease scores compared with controls, indicating further pre-clinical evaluation is necessary before their advancement to clinical trials.

## 1. Introduction

Damage to the central nervous system (CNS) due to injury or disease presents a multi-faceted pathophysiology [1,2]. However, demyelination commonly features and is associated with chronic pain [3,4], incontinence [5,6], cognitive decline [7,8], and paralysis [9,10]. Due to the highly complex nature of CNS pathologies, a combinatorial approach to treatment is likely to be needed. Cellular therapies that promote CNS re/myelination and repair are among the many promising strategies being explored in developing treatment regimens that restore function and quality of life to patients [11,12]. Among the candidate cell therapies being investigated, mesenchymal stromal cells (MSCs) are an attractive option [13,14,15]. Their capacity for immunomodulation and tissue regeneration is well documented [16,17], and their culture requirements are extensively characterized [18,19]. Additionally, they do not necessarily require strict HLA-matching [20,21], with allogeneic MSCs considered preferable in some cases [22,23]. As a result, MSCs are highly amenable to use as cell therapy [24]. Unlike human bone marrow-derived MSCs (hBM-MSCs), which require the invasive and painful aspiration of bone marrow to generate [25], MSCs derived from human olfactory mucosa tissue (hOM-MSCs) can be established from biopsies taken during routine out-patient procedures [26] and are obtained from an anatomical region that experiences neurogenesis throughout life [27]. Transplantation of olfactory ensheathing cells in spinal cord injury patients in a phase I/IIa clinical trial demonstrated no adverse events over the 3-year study period, indicating that cells derived from olfactory mucosa are safe for use in vivo [28]. Additionally, multiple trials of MSC transplantation in spinal cord injury have indicated that these cells are generally well tolerated and safe for treatment of the CNS [13].

We have previously demonstrated that hOM-MSCs are potent mediators of CNS repair in vitro. In mixed cultures of dissociated rodent CNS cells, treatment with conditioned media (CM) from hOM-MSCs resulted in increased myelination levels compared with those seen with hBM-MSC CM [29]. Multiplex analysis of the secretome of hOM-MSCs and hBM-MSCs indicated that while both secreted a wide array of cytokines and chemokines (as is typical of MSCs), CXCL12 was the only analyte found to be significantly higher in hOM-MSC-conditioned media (CM) compared with other cell groups [30]. As well as secreting more CXCL12, hOM-MSCs secreted lower levels of several pro-inflammatory cytokines/chemokines (IL-6, CXCL8, and CCL2) compared to hBM-MSCs. MicroRNA expression analysis indicated that, while sharing 64% identity with hBM-MSCs, 24 were differentially expressed in hOM-MSCs [30], with previous pathway analysis indicating that one of these (miRNA-140-5p) increased expression of CXCL12 [31]. Direct CXCL12 treatment of in vitro myelinating cultures increased myelination, while use of antagomirs of miRNA-140-5p, neutralizing antibodies of CXCL12, or the CXCR4 antagonist AMD3100 all decreased myelination in these cultures (as well as those treated with hOM-MSC CM), confirming CXCL12 as a key secretory factor in this phenotype [30]. hOM-MSCs also demonstrated the capacity to polarize microglia toward an anti-inflammatory phenotype in vitro [30], further indicating potential reparative functions for these cells in the CNS.

We have also demonstrated the efficacy of hOM-MSCs in vivo. In the rat thoracic spinal cord contusion model, injection of hOM-MSCs into the injury site encouraged earlier coordinated gait recovery, with enhanced Schwann cell-mediated remyelination of spared nerve fibers surrounding the injury site [32]. Moreover, in the experimental autoimmune encephalitis (EAE) model, intravenous (IV) injection of hOM-MSCs was seen to reduce disease severity and recovery time compared with control animals and those injected with hBM-MSCs. Furthermore, animals injected with hOM-MSCs saw reduced immune cell infiltrate and axonal abnormalities, as well as improved blood-brain barrier integrity [33].

However, in these studies, hOM-MSCs were isolated and cultured using antibiotics and xeno-derived components. For advancement to clinical trials, these processes must be adapted to meet full GMP compliance. Herein, we describe a GMP-compatible method for the isolation and expansion of hOM-MSCs and evaluate the surface marker expression and secretory profile of the resultant cells. We also interrogate the effects of cryopreservation and different media formulations on cellular phenotypes. Finally, we evaluate these hOM-MSCs in the EAE model to determine efficacy in vivo.

## 2. Results

### 2.1. Determining MSC Identity of GMP-Compatible hOM-MSCs by Surface and Intracellular Marker Expression

Previously, we utilized enzymatic tissue digestion, CD271-positive bead selection, and culture media supplemented with fetal calf serum and antibiotics to establish hOM-MSC cultures [29]. We set out to ascertain if hOM-MSCs could be isolated without reliance on enzymatic digestion or sorting, and instead developed them using explant culture methods and xeno-free GMP-compatible media only. In brief, tissue was mechanically processed (without enzymatic digestion) and cells were allowed to outgrow in GMP-compatible media. Once sufficient outgrowth was observed, cells were harvested, used to establish unsorted cell lines at low and standard density (Unsorted LD/SD), or subjected to the same CD271 bead sort employed previously (CD271 Sort), with the negative fraction/flowthrough (-VE) also seeded at standard density (summarized in Figure 1, full description in materials and methods). For three biopsy samples, processed tissue was split, and the initial outgrowth phase was performed in parallel with and without culture antibiotics. Spent culture media was retained and sent to a fully GMP-accredited external testing center for endotoxin screening. This analysis indicated endotoxin levels were below the limit of detection in all samples tested (lab reports in Appendix C). Consequently, outgrown cells were combined for sorting and passage, and the use of antibiotics was completely dispensed with for the remainder of this study.

MSC identity was confirmed by flow cytometry following initial outgrowth (P0) and over three subsequent passages (P1-3), utilizing a panel of positive (CD73, CD90, CD105) and negative markers (CD14, CD19, CD34, CD45, and HLA-DR), in accordance with International Society for Cell and Gene Therapy (ISCT) criteria for defining MSCs (Figure 2) [34]. Additionally, CD271 was included in this panel due to its use previously in hOM-MSC cell sorting, as was nestin, as it has previously been demonstrated to be highly expressed in hOM-MSCs compared to hBM-MSCs. This analysis indicated that at P0 (Figure 2a(i,ii)), CD90, CD105, and nestin were detectable in >90% of cells, with a higher percentage of the population demonstrating CD73 expression (~98%). At P0, CD271+ cells were present, though they only accounted for around 5% of the total (Figure 2a(iii)), with tSNE analysis indicating at least two distinct populations at this stage (Figure 2a(iv)). However, at P1, the population was effectively 100% positive for CD73, CD90, CD105, and nestin, while CD271 expression was negligible (Figure 2b(i–iii)), and tSNE analysis indicated a single homogenous population (Figure 2b(iv)). This pattern of marker expression remained stable over two subsequent passages for both positive and negative markers (Figure 2c(i,ii)). In order to compare the effects of passaging and sort conditions, a single marker (CD105) was selected. This analysis indicated that while CD105 expression was significantly lower in cells at P0 compared to all subsequent passages (Figure 2d(i)), no notable difference was apparent between sorted and unsorted cells (Figure 2d(ii)). This proved to be the case with all positive markers (summarized in Figure 2d(iii)). These data indicate that CD271 sorting of hOM-MSCs is dispensable, that hOM-MSCs retain MSC identity when isolated and expanded in GMP-compatible conditions (i.e., unsorted LD/SD), and that they retain their characteristic high nestin expression.

### 2.2. Differentiation Capacity of GMP-Compatible hOM-MSCs

According to ISCT criteria, MSC identity is additionally confirmed by their ability to differentiate into a number of other cell types [34]. To evaluate this, GMP-compatible hOM-MSC cell lines from four different donors were subjected to culture conditions that induce adipogenesis (Figure 3a) or osteogenesis (Figure 3b). After a suitable time in culture, cells were either fixed and prepared for staining (i,ii) or for RNA extraction and subsequent qPCR (iii). Oil-red O staining of lipids on untreated controls (Figure 3a(i)) or treated cells (Figure 3a(ii)) clearly indicated typical staining of lipid deposits on the treated cells only. Expression of the adipogenesis-associated genes LPL and PPARγ (Figure 3a(iii,iv)) also indicated considerably higher expression of these genes in treated cells compared with untreated controls. Similarly, Alizarin Red S staining on untreated (Figure 3b(i)) and treated (Figure 3b(ii)) cells demonstrated punctate staining of calcium deposits only in samples where differentiation conditions were applied. Analysis of Osteocalcin (Figure 3b(iii)) and RunX2 (Figure 3b(iv)) also revealed significantly higher expression of these osteogenesis-related genes in the treated samples. Therefore, GMP-compatible hOM-MSCs demonstrate the capacity to differentiate into fat and bone cells.

Collectively, these data indicate that GMP-compatible hOM-MSCs possess both the surface phenotype and differentiation capacity necessary for identification as MSCs by ISCT criteria.

### 2.3. Characterization of the Pro-Myelinating Phenotype of GMP-Compatible hOM-MSCs In Vitro

While demonstrating that hOM-MSCs can be derived in antibiotic- and xeno-component-free conditions is an important initial step in the development of a GMP-compatible cell therapy, it does not address questions of their functional potency. These cells could have an identical flow profile while lacking the pro-myelinating phenotype previously demonstrated by their non-GMP-compliant counterparts. To address this, conditioned media (CM) was produced from passage-matched cells generated by the sort conditions analyzed previously. These were then evaluated in an established myelinating CNS co-culture system alongside unconditioned media (untreated controls) (Figure 4a). The proportion of myelinated axons was enumerated and expressed as a fold change from that seen in ‘Untreated controls’ (represented by the dotted line on the graph, Figure 4b). These data indicated that there was little difference in the capacity of CM from unsorted hOM-MSCs and CD271-sorted cells to promote myelination, and that CM from all tested sort conditions was pro-myelinating. We have previously demonstrated that hOM-MSC secretes a wide array of chemokines and cytokines [30]. However, this work highlighted CXCL12 as specifically upregulated in hOM-MSCs compared with other cell groups and indicated it was likely a key secretory factor in their ability to promote myelination in vitro. Consequently, we considered high CXCL12 to be a critical characteristic for hOM-MSCS to possess. Here, we utilized an ELISA to determine levels of CXCL12 detectable in the CM produced and compared this between sort conditions (Figure 4c). As can be seen, CXCL12 is robustly expressed by hOM-MSCs derived from any sort of condition (~1000 pg/mL per 500 μg whole cell lysate) and is not produced in significantly different quantities between these conditions.

### 2.4. Phenotypic Stability of GMP-Compatible hOM-MSCs Following Cryopreservation

To increase their potential utility as a cell product, we set out to determine if GMP-compatible hOM-MSCs were amenable to cryopreservation without affecting their phenotype. To address this, hOM-MSCs were cryopreserved in a serum-free media formulation readily adaptable to GMP compliance (Knockout Serum Replacement + 10% DMSO), recovered, analyzed by flow cytometry, and CM produced from them (evaluated by ELISA and myelinating cultures). When analyzed by flow cytometry (Figure 5a(i–iii)), a near identical profile of positive surface and intracellular markers as seen prior to cryopreservation is demonstrated, with cryopreserved cells retaining essentially 100% expression of CD73, CD90, CD105, and nestin and remaining CD271 negative. Indeed, when looking at both positive and negative marker expression (Figure 5b), cryopreserved hOM-MSCs clearly retain a marker profile compatible with ISCT criteria. As mentioned, CXCL12 production is considered critical to hOM-MSCs ability to promote myelination. To ensure this capacity of hOM-MSCs is not compromised by cryopreservation of the cells, CM was generated as before from recovered cells, and levels of CXCL12 in this CM were determined by ELISA (Figure 5c). Here we see that CXCL12 production remains comparable to that seen in freshly isolated cells. To confirm if this correlated with a functional endpoint, further myelinating cultures were conducted using this CM, with myelinated axons enumerated as before (expressed as fold change from untreated controls, represented by a dotted line at 1) and compared with the level achieved previously with CM from freshly isolated cells (Figure 5d,e). As was seen previously, CM from both freshly isolated and cryopreserved cells is pro-myelinating, with no significant differences noted between them. Therefore, these data would indicate that hOM-MSCs are not phenotypically altered by cryopreservation and thawing.

### 2.5. Comparing hOM-MSCs Expanded with GMP-Compliant and Xeno-Derived Media Supplements

To determine if the change in medium formulationfrom utilizing xeno-derived components to GMP-compliant alternatives may have implications for their clinical efficacy, hOM-MSC lines were grown in either the GMP-compatible medium formulation used so far (referred to as 5% nLiven) or one supplemented instead with 10% fetal calf serum (referred to as 10% FCS). These were then compared by flow cytometry (Figure 6a,b). Media used had no effect on positive marker expression, with both groups demonstrating > 99% positivity for CD73, CD90, CD105, and nestin. Interestingly, there are marginal (but significant) differences in staining for several negative markers (5% nLiven vs. 10% FCS; CD14 = 0.052/0.822, CD34 = 0.048/0.392, CD45 = 0.048/0.596). Notably, these are all far below the 5% cut-off for expression of these markers stipulated by the ISCT, indicating that changing media did not alter MSC identity. As before, CM was generated from each group, evaluated for CXCL12 levels by ELISA (Figure 6c), and used in myelinating cultures (Figure 6d,e). Again, in both assays, changing media demonstrated no significant differences, producing broadly comparable levels of CXCL12 (213.1/321.2 pg/mL, normalized to BCA) and similar levels of myelination (1.69/1.38 fold change from controls). Therefore, the change to GMP-compatible media has not meaningfully altered the hOM-MSC phenotype in these assays.

### 2.6. Evaluating In Vivo Efficacy of GMP-Compatible hOM-MSCs in the EAE Model

We sought to determine if GMP-compatible hOM-MSCs would be effective in ameliorating EAE severity and improving recovery time. PBS (vehicle control) or 1 × 10^6^ GMP-compatible hOM-MSCs were administered soon after disease onset, and scores and weight were monitored until the end of the clinical course. Surprisingly, these data indicated that GMP-compatible hOM-MSCs had little capacity to ameliorate disease severity (Figure 7a(i) and had no overt impact on recovery of weight (Figure 7a(ii)). When comparing scores (Figure 7b(i)) or weight (Figure 7b(ii)) on day 32 of, or the maximum score seen during the clinical course (Figure 7b(iii)), we see little difference between animals treated with PBS or GMP-compatible hOM-MSCs (PBS/5% nLiven; b(i) D32 EAE score = 2.217/1.958, b(ii) D32 Weight = 20.47/19.98 g, b(iii) Max. EAE score = 3.70/3.64). To determine if there were any changes in speed of recovery, linear regression analysis was performed on score data from 1 day post-treatment until the end of the model (Figure 7c(i)), and the total area under the curve was determined for every animal in each group (Figure 7c(ii)). However, this analysis again indicates that treatment with GMP-compatible hOM-MSCs did not significantly reduce recovery time compared with control animals. (PBS/5% nLiven; AUC = 57.14/54.45). Therefore, it appears that while pro-myelinating in vitro, GMP-compatible hOM-MSCs are not able to demonstrate significant in vivo efficacy in this model of demyelinating disease.

## 3. Discussion

We describe in this study a standardized and GMP-compatible protocol for the isolation and expansion of hOM-MSCs. In the future, achieving a fully GMP-compliant cell product for use in humans would require the use of appropriate manufacturing facilities. However, by completely removing our reliance on xeno-derived components in the culture medium and refining our sample preparation and culture methods to eliminate microbial contamination (without relying on antibiotics), we are confident that we have developed a process for the establishment of hOM-MSCs that is readily translatable to full GMP compliance. These cells have been thoroughly characterized, conform to ISCT criteria for identification as MSCs [34], demonstrate high expression of nestin and CXCL12 (a previously established trait of hOM-MSCs [29,30,33]), and so we would propose these as key release criteria for any future cell product developed from this method. Additionally, in 3/3 samples tested in a GMP-compliant QC lab, endotoxin levels within spent culture media from these cells were below the limit of detection for the assay (Appendix C). Given that the ultimate end goal for such a product would be the treatment of CNS injury and disease (an immune-privileged anatomical region), we would also suggest that an acceptable endotoxin release level of effectively zero be used for future release criteria.

Previous methods of generating hOM-MSCs utilized non-GMP-compliant CD271 bead sorting [29,30]. We have demonstrated elsewhere that CD271-sorted cells have superior pro-myelinating and neuroprotective properties compared with cells grown from the negative sort fraction and hBM-MSCs [29,30,32,33]. However, MSCs isolated from other tissues often do not require sorting and instead are developed into a homogenous population by culture methods alone [35,36]. Given that hOM-MSC has previously demonstrated favorable growth kinetics, we reasoned that such an approach might work here. When analyzed by flow cytometry, pre-sort cells (P0) already demonstrated a predominance of MSC-like cells, with >90% being CD90, CD105, and nestin positive (near 100% for CD73). Interestingly, only a small percentage of P0 cells were positive for CD271 expression (~5%). Moreover, within this CD271+ population, only a smaller proportion (around 15% on average) expressed CD90, CD105, or nestin, with around 80% expressing CD73 (Appendix B, Figure A1), indicating not all outgrown CD271+ cells from nasal biopsies were MSCs. However, MSC marker expression was effectively uniform within one passage for both sorted and unsorted cell lines, with CD271 being undetectable from P1 onward (including in CD271-sorted cell lines). Additionally, unsorted cells were equivalent to sorted counterparts in their capacity to produce CXCL12 and promote myelination in vitro, further confirming that sorting is not required to generate cells with favorable characteristics. Collectively, these data would appear to indicate that CD271 positive selection may not be required to obtain a pure population of hOM-MSCs, and that cell lines established without sorting are phenotypically equivalent to those that are.

Enzymatic digestion was previously utilized in our tissue processing protocol [29,30], and was also recently used in the development of a GMP-compliant method for the isolation of olfactory ecto-mesenchymal stem cells (analogous to hOM-MSCs), where it was found to improve initial cell yield and reduce contamination [37]. However, other investigations conflict with this, reporting that enzymatic digestion can negatively impact initial MSC recovery, viability, surface marker expression, and growth factor production and increase heterogeneity in the initial cell population [38,39,40]. Considering this and our own flow cytometry results indicating > 90% MSC identity of initially outgrown cells, the decision was made to take a purely explant-based approach to tissue processing throughout this study.

As well as substantially advancing hOM-MSCs toward GMP compliance, this explant and passaging method is also simpler, cheaper, and bypasses the need to develop GMP-compliant sorting methods (with expensive associated reagents). However, it does infer that, in previous studies, the sorting of hOM-MSCs mostly served to debulk the starting population to a far smaller number from which homogenous hOM-MSCs grew out. By starting with a comparatively sparse starting density of unsorted cells, we may, in essence, be replicating this effect. This would also be advantageous in expanding hOM-MSCs to clinically relevant quantities quickly, however, as a larger proportion (if not all) of the initial cellular outgrowth from a tissue sample could be used to establish a cell line rather than a relative few sorted cells.

The ability to be cryopreserved and recovered without significant loss of potency or viability greatly increases the utility of cell therapies, allowing for longer-term storage and the potential to establish cell banks for more rapid treatment [41,42]. Given that earlier intervention correlates with better recovery of function in conditions such as spinal cord injury [13,43], establishing that GMP-compatible hOM-MSCs are not negatively affected by cryopreservation and recovery is critical. We have demonstrated here that hOM-MSCs are highly amenable to cryopreservation, with no significant differences in phenotypic marker expression, CXCL12 production, or promotion of myelination compared with fresh cells. Therefore, GMP-compatible hOM-MSCs can be cryopreserved without altering their phenotype or compromising their favorable characteristics. In comparison, the reported effects of cryopreservation of hBM-MSCs show considerable variability, with a wide range of post-thaw viability described and some reports of attenuated paracrine effects [44]. Consequently, hOM-MSCs may be more suitable for cryopreservation and ‘banking’ than hBM-MSCs.

Previous work utilized αMEM supplemented with 10% FCS to culture hOM-MSCs [29,30]. In this study, we have again used αMEM (as GMP-compliant variants are widely available and are identical in composition) but switched to a GMP-compliant human platelet lysate supplement (nLiven). While we have found hOM-MSCs thrive in this medium, changes in culture media have previously been observed to induce phenotypic changes in MSCs [19,45,46]. However, when grown in parallel in either 10% FCS-supplemented or GMP-compatible media, hOM-MSCS demonstrated no significant changes in observed phenotype, indicating that the transition to GMP-compliant alternatives has not diminished their potential therapeutic value.

To evaluate GMP-compatible hOM-MSCs in vivo, the EAE model was selected, as hOM-MSCs have previously demonstrated potency in this model of immune-mediated demyelination [33]. While treatment with hOM-MSCs did not cause any adverse events during the clinical course and demonstrated no enhanced lethality (Appendix B, Figure A2), it did not significantly improve disease outcomes. Differences between treated and control animals were minor and did not meet the threshold for statistical significance over several measures. This observation was surprising, given our in vitro data. However, this could be attributable to several factors. Firstly, disease severity was higher than anticipated in these experiments (particularly in animals immunized with full-length MOG protein), with a number of animals culled on welfare grounds. It is conceivable that in such a runaway inflammatory environment, hOM-MSCs are insufficient to curtail disease. Cells grown in 10% FCS did demonstrate some minor (but significant) changes in the expression of negative MSC markers compared with those grown in GMP-compliant media. However, critically, none approached the 5% threshold for these markers stipulated by the ISCT, so the biological significance of these differences is unclear. Additionally, cells grown in 10% FCS did trend toward slightly higher CXCL12 production than those grown in GMP-compliant media, although not significantly so, and so it is unclear if this would account for the lack of efficacy of GMP-compatible cells in vivo. Less overt phenotypic changes introduced by the change to GMP-compliant media cannot be detected in these assays and may require more in-depth transcriptomics to discern. It is also important to highlight that while we have evaluated these cells in both the historic and new media formulations, we did change the supplier of FCS from that used previously (due to issues of availability). Although unlikely to be a sole factor for observed differences from our prior findings, this, coupled with CD271 selection, may have introduced subtle phenotypic differences that were not observable in our in vitro characterization of these cells. Therefore, further pre-clinical work is warranted, likely in alternative models of demyelinating disease (such as the spinal cord contusion model).

Regardless, we demonstrate here that hOM-MSCs can be isolated and expanded using techniques and reagents that are readily translatable to full GMP compliance. These cells conform to ISCT minimal criteria for identification as MSCs, can be cryopreserved without phenotypic change, promote myelination in vitro, and are safe for intravenous delivery in vivo. As such, we would suggest further development of GMP-compatible hOM-MSCs as a potentially valuable asset in developing multi-faceted treatment strategies for demyelinating conditions in the CNS.

## 4. Materials and Methods

### 4.1. Nasal Biopsy Processing and hOM-MSC Cell Line Generation

Biopsies were obtained with Central Office for Research Ethics Committees (COREC, REC reference 07/S0710/24) ethical approval and informed patient consent from six patients undergoing routine nasal septoplasty/polypectomy surgery (in accordance with the requirements of the Declaration of Helsinki). Tissue was taken from the nasal and sinus mucosa (primarily from ethmoid sinus mucosa), from which we have previously shown mesenchymal stromal cells (hOM-MSCs) that promote CNS repair can be derived [29,30]. Biopsies were collected on ice in Hanks balanced salt solution (HBSS) containing penicillin (100 units/mL), streptomycin (100 μg/mL), and fungizone (Amphotericin B, 1.25 μg/mL) (all Thermo Fisher Scientific, Waltham, MA, USA) for transport to our tissue culture facilities. In a biological safety cabinet (BSC) and under sterile conditions, tissue was placed in a Petri dish and extraneous bone and non-mucosa tissue removed. All subsequent steps were also conducted in a BSC under sterile conditions. The remaining tissue was ‘washed’ by swirling in a Petri containing 70% ethanol for 30–60 s, then ‘rinsed’ in the same manner in PBS (Thermo Fisher Scientific, Waltham, MA, USA, step performed twice) to remove residual alcohol. Tissue was then transferred to an Eppendorf containing 1 mL of culture media (αMEM with nucleosides and Glutamax (Thermo Fisher Scientific, Waltham, MA, USA), supplemented with 5% nLiven PR GMP-ready human platelet lysate (Sexton Biotechnologies, Indianapolis, IN, USA), henceforth referred to as ‘MSC media’ and used throughout this study unless otherwise stated). Using straight spring-bow surgical scissors, tissue was then minced until no pieces larger than 1–2 mm remained. The processed sample was transferred to a 10 cm TC-treated culture dish (Corning, New York, NY, USA) containing MSC media, which was then placed in a humidified incubator (set to 37 °C, 5% CO_2_) and left undisturbed for 5–7 days. By this time, some tissue had adhered to the culture dish, and small clusters of outgrowing cells were apparent. Spent media and unadhered tissue were removed, and the adherent tissue and cells were washed gently with PBS. MSC media was replaced, the dish returned to the incubator, and the media was changed every 5 days. For endotoxin testing, the initial processed tissue was split between two dishes and grown in parallel in either MSC media or media further supplemented with antibiotics (1% penicillin/streptomycin solution, Sigma Aldrich, St. Louis, MO, USA). Spent media was reserved and sent for analysis at a clinical-grade QC laboratory for endotoxin testing. This testing indicated no detectable presence of endotoxin in any sample tested (Appendix C).

Once large cell clusters were present and beginning to merge (~10 days), these were considered ready for further processing. Cell monolayers were washed twice with PBS, then incubated in TrypLE Select dissociation buffer (Thermo Fisher Scientific, Waltham, MA, USA), returned to the incubator, and agitated every few minutes until cells began to lift off. TrypLE was then neutralized with a 4× volume of culture media, and cell clumps were broken up by trituration. The cell suspension was passed through a 70 µm cell filter (Greiner, Kremsmünster, Austria), pelleted, and resuspended in PBS, with cell counts taken. Where possible, unsorted cells from this initial out-growth phase (considered passage 0/P0) were reserved for analysis by flow cytometry.

Cell lines were established from outgrown cells in four different ways. Unsorted cells were used to establish cell lines by initially seeding them at either 500–750 cells/cm^2^ (unsorted low density, Uns LD) or 3500 cells/cm^2^ (unsorted standard density, Uns SD) in MSC media on TC-treated culture flasks (Corning, New York, NY, USA). Sorting of hOM-MSCs was also performed using a CD271 positive selection kit (StemCell Technologies, Vancouver, Canada) as described previously. In brief, cells were resuspended in 100 µL sorting buffer (PBS +0.5% bovine serum albumin (BSA, Sigma Aldrich, St. Louis, MO, USA) and 2 mM EDTA, Thermo Fisher Scientific, Waltham, MA, USA, henceforth referred to as PEB), and transferred to a 5 mL round bottom polystyrene tube (Corning, New York, NY, USA). Here, 2.5 µL FcR blocker and 5 µL antibody cocktail per sample were added to this, with the sample mixed and incubated for 15 min at room temperature. Then, 5 µL of RapidSpheres was added to each sample, mixed, incubated for a further 5 min, and then brought up to a volume of 2.5 mL with PEB. The tube was placed in the sorting magnet for 5 min, the supernatant was transferred to a clean tube, and cells were resuspended in 2.5 mL PEB. This step was repeated, and the remaining sorted cells were washed once in PBS and resuspended in 0.25 mL of MSC media. These were seeded in a single spot in a TC-treated flask and placed in the incubator for 2 h to allow cells to attach, after which the full volume of MSC media was added to the flask (CD271 sorted). Flowthrough cells from the sort (i.e., the negative fraction) were washed with PBS, resuspended in MSC media, cell counts taken, and seeded at 3500 cells/cm^2^ in TC-treated flasks (–VE) (summarized in Figure 1). Also, 75 cm^2^ tissue culture flasks (Corning, New York, NY, USA) were used for maintaining cultures throughout. The only exception was with CD271-sorted cells, which were cultured initially in 25 cm^2^ flasks (Corning, New York, NY, USA) at P0 due to low cell numbers and in 75 cm^2^ for subsequent passages.

### 4.2. Cell Culture

Once established, all new cell lines were maintained in the same manner, and grown in MSC media on TC-treated plastic (Corning, New York, NY, USA). Once reaching 70–90% confluence, cell monolayers were washed twice with PBS, incubated with a volume of TrypLE sufficient to cover the cell layer, and returned to the incubator. Cells were occasionally agitated until lifted from the plastic, TrypLE was neutralized using 4× volume of MSC media, and a monocellular suspension was created by trituration by pipette. Cells were pelleted (300× *g* for 5 min), supernatant discarded, and cells resuspended in an appropriate volume of MSC media. Cell counts were performed, and new flasks were seeded at 3500 cells/cm^2^, with unused cells retained for further analysis. These were passaged a total of three times, with aliquots of cells at each passage cryopreserved wherever possible (see below). Where indicated, hOM-MSCs were also grown in αMEM supplemented with 10% FCS (Sigma Aldrich, St. Louis, MO, USA), as per their previous culture conditions.

### 4.3. Flow Cytometry and Antibody Panel

Single suspensions of hOM-MSCs were created as described above, resuspended in PBS, and cell counts taken. The preparation of the flow samples was performed as described in [47]. In brief, aliquots of 2–3 × 10^5^ cells per sample (consistent within each run) were transferred to a 96-well round bottom plate (Corning, New York, NY, USA), with residual cells pooled and used for setting up controls. From this point, all subsequent steps were performed while protected from the light, and at 4 °C unless specifically stated otherwise. Cells were washed in PBS, then stained with fixable viability stain (Thermo Fisher Scientific, Waltham, MA, USA, 1:1000 in PBS) for 15 min. Cells were washed in PEB, then incubated with a human FcR blocking agent (Miltenyi Biotec, Bergisch Gladbach, Germany, 1 µL per sample in 200 µL PEB) for 10 min. Samples were then resuspended in 50 µL of the full surface marker antibody staining cocktail or a fluorescence minus one control (where a single antibody is excluded as a negative control for that stain) in Brilliant Stain buffer (BD Biosciences, Franklin Lakes, NJ, USA, antibody cocktail detailed in Table 1) for 20 min. Antibodies were obtained for each marker from the manufacturer indicated in Table 1 (either Biolegend, San Diego, CA, USA or BD Biosciences, Franklin Lakes, NJ, USA). Samples were then washed with PEB and prepared and stained for intracellular markers (nestin; see Table 1) using the eBiosciences Intracellular Fixation and Permeabilization kit (Thermo Fisher Scientific, Waltham, MA, USA), according to kit instructions. Samples were analyzed using a BD LSR Fortessa analyzer (BD Biosciences, Franklin Lakes, NJ, USA, based in the School of Immunity and Inflammation’s Flow Cytometry Core Facility), and the resulting data were analyzed using FlowJo v10 software (BD Biosciences, Franklin Lakes, NJ, USA). Flow cytometry data is expressed as the percentage (%) of live cells, unless otherwise stated. Gating strategies are described in Appendix B, Figure A3.

### 4.4. Adipogenic/Osteogenic Differentiation

A selection of hOM-MSCs (Uns SD, passage 3) were selected for use with either the StemPro Adipogenesis or Osteogenesis Differentiation Kits (Thermo Fisher Scientific, Waltham, MA, USA), in accordance with the manufacturer’s instructions with slight modifications. In brief, in both cases, hOM-MSCs were seeded into either 6-well (Corning, New York, NY, USA) or 48-well (StarLab, Milton Keynes, UK, seeded in triplicate) plates at a density of 1 × 10^4^ cells/mL in MSC medium. Upon reaching 100% (adipogenesis) or 70% (osteogenesis) confluency, these were switched over to differentiation media, which was subsequently refreshed every 3–4 days. At either 28 (adipogenesis) or 42 (osteogenesis) days, cells were either stained for lipid/calcium deposition (48-well) or harvested for RNA (6-well).

### 4.5. Oil Red O/Alizarin Red Staining

Differentiated cultures were fixed by removing differentiation media, washing cell layers twice with PBS, and being fixed in 4% paraformaldehyde (PFA, Sigma Aldrich, St. Louis, MO, USA) in PBS for 20 min at room temperature. PFA was then removed, and cultures were washed twice with PBS. Oil Red O staining of lipid deposits was performed as previously described. In brief, PBS was removed, cells were washed ×1 with distilled water and ×1 with 60% isopropanol (Sigma Aldrich, St. Louis, MO), and they were incubated in a 0.3% Oil Red O (Sigma Aldrich, St. Louis, MO, USA) solution in 60% isopropanol for 15 min at room temperature. The staining solution was then removed, and cells were washed several times with 60% isopropanol until the non-bound stain could no longer be removed. Alizarin Red S staining for calcium deposition was also performed as described by others. In brief, PBS was removed, cells were washed twice with distilled water, and Alizarin Red S (Sigma Aldrich, St. Louis, MO, USA) staining solution pH 4.2 sufficient to cover the cell layers was added to each well and incubated at room temperature for 5 min, protected from light. Each well was then rinsed 3× with distilled water. In both cases, distilled water was added to cultures after final wash steps to prevent dehydration. These were imaged with a Zeiss Primovert light microscope at 400× magnification, with images captured using the AxioCam ERc5s camera using Zeiss Zen 2.3 (blue edition) software.

### 4.6. RNA Extractions, cDNA Conversions and qPCR Analysis

Primer sequences can be found in Table 2. Genes of interest to confirm adipo/osteogenic differentiation were selected [48] primer pairs were designed to span exon junctions using Primer3 [49]. RNA extraction, cDNA conversion, and qPCR were performed as described [50]. In brief, RNA was extracted from basal and differentiated hOM-MSCs (PureLink RNA Mini Kit, Thermo Fisher Scientific, Waltham, MA, USA), with on-column DNase digestion performed (RNase-Free DNase set, Qiagen, Hilden, Germany) to remove contaminating genomic DNA. RNA concentration was determined using the DeNovix DS-11+ spectrophotometer (DeNovix, Wilmington, DE, USA), and 0.5/1 μg of RNA was converted to cDNA (High-Capacity RNA-to-cDNA kit, Thermo Fisher Scientific, Waltham, MA, USA). The resulting cDNA was diluted in nuclease-free water (Qiagen, Hilden, Germany, diluted 1/5 or 1/10 depending on the starting RNA quantity) for use in qPCR reactions. qPCR reactions were prepared in 384-well plates using PerfeCTa SYBR Green FastMix ROX (Quantabio, Beverly, MA, USA), a final primer concentration of 0.75 μM, 1 μL cDNA template per reaction, and a total volume of 10 μL with nuclease-free water. Plates were run on the QuantStudio 7 Flex system (Thermo Fisher Scientific, Waltham, MA, USA), with results analyzed and exported to spreadsheets using the onboard software. Data were expressed relative to housekeeping and undifferentiated controls using the 2^−ΔΔCT^ method.

### 4.7. Conditioned Media and Myelinating Cultures

Conditioned media was prepared as follows: hOM-MSC at passage 3 was grown to 100% confluency in MSC media in 75 cm^2^ TC flasks (Corning, New York, NY, USA). At this point, spent media was aspirated, cells washed once with PBS, and media replaced with 11 mL DMEM-media (DMEM 4.5 g/L glucose, Invitrogen, 50 nM hydrocortisone, 0.5% N1 medium supplement, and 10 ng/mL biotin, all Sigma Aldrich, St. Louis, MO, USA). These cultures were left for 72 h, after which DMEM- was removed, filtered through a 0.2 µm syringe filter (Sigma Aldrich, St. Louis, MO, USA), and stored in a −80 °C freezer until required. When prepared for myelinating cultures, this was diluted 1:2–4 with fresh DMEM-.

Myelinating cultures were set up as described previously [51]. Briefly, neurospheres generated using striata from 1-day-old Sprague-Dawley (SD) rats, triturated by pipette, were seeded onto poly-lysine-coated (using a 13 μg/mL solution, Sigma Aldrich, St. Louis, MO, USA) 13 mm coverslips (VWR), with 3 coverslips per 35 mm Petri dish (Corning, New York, NY, USA). These were incubated for 5–7 days in DMEM (1 g/mL glucose, Thermo Fisher Scientific, Waltham, MA, USA) supplemented with 10% FCS (Sigma Aldrich, St. Louis, MO, USA). Once complete monolayers of astrocytes were apparent, these were used as support cells to set up mixed-cell myelinating cultures. For these, spinal cords were dissected from E15.5 SD spinal cords and subsequently digested with a mixture of collagenase I (1.33% *w*/*v* in L-15 media, both Thermo Fisher Scientific, Waltham, MA, USA) and 0.25% Trypsin-EDTA (Sigma Aldrich, St. Louis, MO, USA) for 15 min at 37 °C. Reactions were quenched with a mix of 0.52 mg/mL soybean trypsin inhibitor, 3 μg/mL BSA fraction V, and 0.04 mg/mL DNase (all Sigma Aldrich, St. Louis, MO, USA), again prepared in L-15 media. The tissue mixture was pelleted (at 800 rpm for 5 min), and cells were resuspended in plating media (50% *v*/*v* DMEM, 1 g/mL glucose, 25% *v*/*v* horse serum, 25% *v*/*v* HBSS, and 2 mM L-glutamine, all Thermo Fisher Scientific, Waltham, MA, USA).

Cell counts were taken, volume adjusted to give a density of 1.5 × 10^6^ cells/mL, and 100 μL of this suspension seeded directly on to the coverslips. These were placed in the incubator for 2 h to allow cells to adhere, after which they were supplemented with an additional 300 μL of plating media and 500 μL of DMEM+ (same formulation as DMEM-, with the addition of 0.5 mg/mL insulin, Sigma Aldrich, St. Louis, MO, USA). Media was replenished by removing 400 μL and adding 500 μL DMEM+ (to account for evaporation) every 2–3 days up to 12 days. From days 12–28, media was replenished with either basal DMEM- or the previously mentioned CM/DMEM- mix, again replenished every 2–3 days. After 28 days, media was removed, cultures washed twice with PBS, and fixed with 4% paraformaldehyde (PFA, Sigma Aldrich, St. Louis, MO, USA) for 20 min at room temperature. PFA was removed, cultures washed twice with PBS, and a volume of PBS sufficient to comfortably cover the cell layers was added for storage at 4 °C until required for staining and imaging (plates sealed with parafilm to prevent evaporation).

### 4.8. Immunocytochemistry

To immunolabel the cells, they were first permeabilized with 0.2% Triton X-100 (Sigma Aldrich, St. Louis, MO, USA) at room temperature (RT) for 15 min, and blocked with phosphate-buffered saline (PBS) with 0.2% porcine gelatin (blocking buffer, Sigma Aldrich, St. Louis, MO, USA) for 1 h at RT. The primary antibodies were diluted in a blocking buffer, and the cells were incubated for 1 h at RT. Mature myelin (proteolipid protein, PLP) was visualized using the AA3 antibody (1:100, anti-rat; hybridoma supernatant gift [52]), and neurofilament was detected using SMI31 (mouse IgG1, 1:1500, Biolegend, San Diego, CA, USA). After washing, the cultures were incubated with the appropriate secondary antibodies for 45 min at RT and mounted under coverslips using Vectashield anti-fade (Vector Laboratories, Peterborough, UK).

### 4.9. Microscopy and Image Analysis

Quantification of myelination was carried out by collecting images at 10× magnification using an Olympus BX51 (Olympus, Essex, UK). For each coverslip, 10 images were taken randomly, covering the entire coverslip. In each biological repeat, coverslips were stained in triplicate; thus, 30 images were taken per condition (at least n = 3 biological repeats). The analysis of treated cultures was carried out by comparing them to non-treated control cultures run in parallel. Analysis was carried out using Cell Profiler Image Analysis software (Broad Institute, v1.3) and is available to download at https://github.com/muecs/cp (accessed 8 May 2022). This uses pattern recognition software to distinguish between linear myelinated internodes and oligodendrocyte cell bodies. In this manner, we track the co-expression of myelin sheaths (PLP) and axons (SMI31), which ignores immature oligodendrocytes lacking axonal contact, and therefore allows us to calculate the percentage of myelinated fibers. Once enumerated, samples were normalized to unconditioned media-treated controls, and values were expressed as fold changes from these controls.

### 4.10. Protein Extractions and ELISA Analysis

Whole-cell lysate (WCL) was prepared from cells used in the creation of CM. Cells were lifted with TrypLE as described previously, pelleted, and washed once in PBS, then resuspended in M-PER Mammalian Protein Extraction Reagent (Thermo Fisher Scientific, Waltham, MA, USA) and vortexed every 1–2 min for 15 min (with samples kept on ice when not being vortexed). Samples were then centrifuged (>10,000 RPM, 4 °C, 15 min) and supernatants transferred to clean Eppendorf’s and either kept on ice or stored at −20 °C for future analysis. Protein concentration was determined by a BCA assay (Thermo Fisher Scientific, Waltham, MA, USA) and used to normalize ELISA data. ELISA analysis for CXCL12 was conducted on the same CM used for myelinating cultures, using the Human SDF-1α (CXCL12) Mini ABTS ELISA Development Kit and associated ancillary reagents (Peprotech, now part of Thermo Fisher Scientific, Waltham, MA, USA). Plates were read on the Varioskan LUX plate reader (Thermo Fisher Scientific, Waltham, MA, USA, read at 405 nm absorbance with correction at 650 nm) over various timepoints. The concentration of CXCL12 in CM was calculated against the provided standard and expressed relative to 200/500 μg of WCL.

### 4.11. Cryopreservation and Recovery

‘Freezing media’ (FM) was prepared as follows: 90% *v*/*v* Knockout Serum Replacement (Thermo Fisher Scientific, Waltham, MA, USA), 10% *v*/*v* DMSO (Thermo Fisher Scientific, Waltham, MA, USA). Prior to cryopreservation, cells were prepared as a single-celled suspension as described already, resuspended in PBS, and cell counts taken. Cells were then resuspended at a density of 8 × 10^6^ cells/mL in FM, and 2 × 10^6^ cells were transferred to cryovials (Simport Sceintific, Saint-Mathieu-de-Beloeil, Canada). These were then slowly cooled (~−1 °C/minute) using a ‘Mr Frosty’ freezing container (Thermo Fisher Scientific, Waltham, MA, USA) in a −80 °C freezer before being transferred to liquid nitrogen tanks for long-term storage. For recovery, cells were thawed quickly in pre-warmed culture media, pelleted, resuspended in fresh culture media, and counted. They were then seeded in TC flasks at a density of 3500/cm^2^. The media was refreshed after 24 h to remove any residual DMSO that may have leached out of the cells and to remove any dead cells. From this point, these cells were then cultured as described previously.

### 4.12. EAE Induction

A total of 33 female (SL ADD) C57Bl/6 J mice were purchased from Harlan Laboratories (Loughborough, UK). All mice were housed under a 12 h light/dark cycle with ad libitum access to food and water in pathogen-free conditions. All experimental procedures were performed in accordance with the UK Animals (Scientific Procedures) Act 1986. All applicable international, national, and/or institutional guidelines for the care and use of animals were followed. The research protocol was approved by the Ethical Committee for Animal Experimentation at the University of Glasgow, UK.

EAE was induced in female mice (7–8 weeks of age, weighing 18.5 ± 1.5 g) by subcutaneous injection at one site at the tail base with an emulsion (100 µL total) containing 200 µg recombinant rat myelin oligodendrocyte glycoprotein protein spanning amino acids 1-125 (MOG1-125) in complete Freund’s adjuvant (Sigma Aldrich, St. Louis, MO, USA) supplemented with 300 µg Mycobacterium tuberculosis (strain H37RA; BD Biosciences, Franklin Lakes, NJ, USA) [34]. In some cases, mice were instead immunized with myelin oligodendrocyte glycoprotein peptide spanning amino acids 33–55 (MOG33-55) (SynPeptide, Shanghai, China) in complete Freund’s adjuvant (Sigma Aldrich, St. Louis, MO, USA) supplemented with 300 μg Mycobacterium tuberculosis (strain H37RA; BD Biosciences, Franklin Lakes, NJ, USA). Mice were then injected intraperitoneally with 200 ng pertussis toxin (Enzo Life Sciences, Farmingdale, NY, USA) in 100 µL of phosphate buffered saline solution (PBS, pH 7.6) immediately and 48 h after the immunization. The mice were scored daily for clinical manifestations of EAE on a half-point scale of 0–5. hOM-MSCs (1 × 10^6^ cells in 100 μL PBS) or PBS (100 μL) were administered when animals showed signs of clinical disease (score of at least 2; hind limb paralysis) by intravenous (IV) injection of the tail vein. Mice were randomly distributed between groups on the day of IV administration and scored daily blind. This treatment strategy prevented the inclusion of asymptomatic animals.

### 4.13. Statistical Analysis

All statistical analysis performed in this study (and generation of graphical representations thereof) was carried out using the GraphPad Prism statistical software package (versions 9.3.1, GraphPad Software, Boston, MA, USA). The specific statistical tests employed are indicated in the relevant figure legends.

## 5. Conclusions

MSCs derived from olfactory mucosa tissue are highly amenable to culture in GMP-compliant media formulations and can be grown to a homogenous population, demonstrating MSC surface marker phenotype in one passage without cell sorting. Like MSCs derived from other tissues, they are able to differentiate into fat and bone cells, and their properties remain stable following cryopreservation. The change in media formulation does not affect their surface marker expression, CXCL12 production, or pro-myelinating properties. However, when tested in vivo, these near-GMP-compliant cells do not appreciably ameliorate disease in the EAE model, indicating further pre-clinical investigation is required.

## Figures and Tables

**Figure 1 ijms-25-00743-f001:**
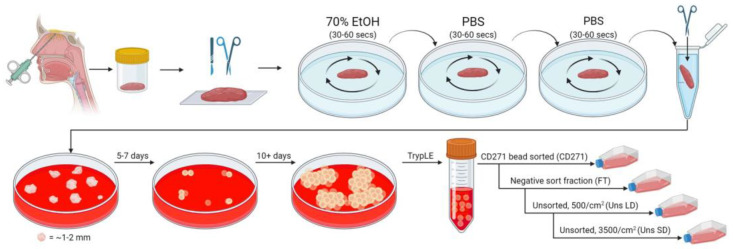
Graphical summary of hOM-MSC isolation method. Biopsy samples from the olfactory mucosa were processed by removing non-mucosal tissue, then washed first in 70% ethanol, and twice in PBS. The tissue was then minced and transferred to tissue culture plastics with culture media until colonies of outgrown cells emerged. Cell cultures were established from unsorted cells seeded at 500–750 or 3500 cells/cm^2^, or CD271 bead sorted and established from the sorted or negative fractions (also at 3500/cm^2^ for the latter).

**Figure 2 ijms-25-00743-f002:**
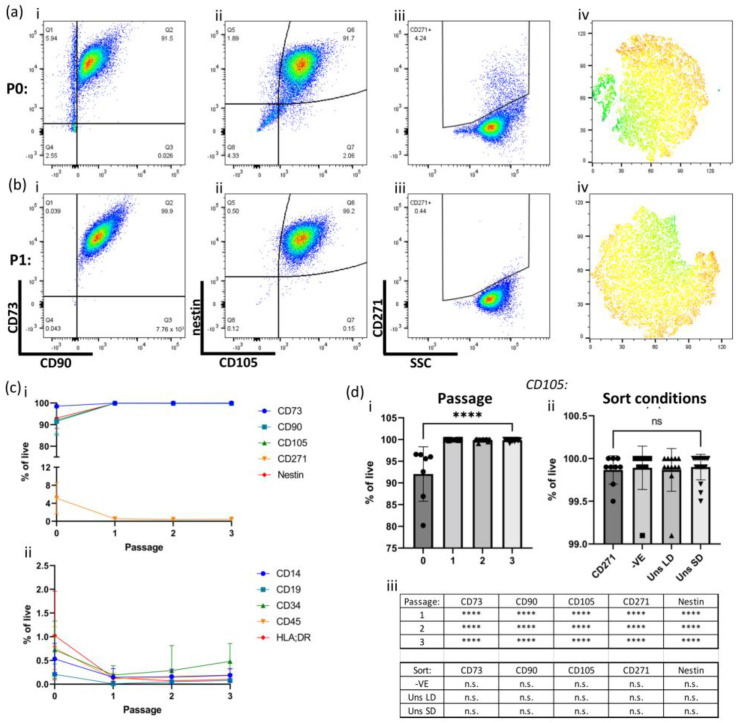
Sorting of hOM-MSCs is dispensable, with cells demonstrating an MSC profile within one passage. Cells from the initial out-growth phase (**a**) and after one passage (**b**) were evaluated by flow cytometry, with exemplar plots shown. The same cells (pre-gated on live singlets) were compared for expression of CD73, CD90 (**a**,**b**(**i**)), nestin, CD105 (**a**,**b**(**ii**)) and CD271 (**a**,**b**(**iii**)). (**a**,**b**(**iv**)) tSNE analysis was also performed to determine homogeneity within the population sample, colored by levels of nestin expression (blue/green = low expression, yellow/orange/red = high expression). (**c**) The percentage of live cells expressing the ‘positive’ (**c**(**i**)) and ‘negative’ (**c**(**ii**)) MSC markers was tracked over passages 0-3, presented as the mean ± SD of all available samples at each passage (5/6 donors for P0, 6/6 for all subsequent passages). Selecting one marker for comparison (**d**), expression of CD105 was evaluated across the first 3 passages (**d**(**i**)) or between sorted and unsorted cells at passage 1 (**d**(**ii**)). This same analysis was performed for all positive MSC markers and summarized, with a statistical comparison made between P0 and all subsequent passages (top) and CD271-sorted cells and all other sort conditions (bottom) (**d**(**iii**)). Data represent six independent donors, with individual data points assigned as either (**d**(**i**)) different sort conditions per donor (CD271), bead sorted (CD271), the negative fraction flowthrough from the sort -VE), unsorted cells seeded at low density (Uns LD), or standard density (Uns SD) across multiple passages, or (**d**(**ii**)) different passages of the same sort conditions. Data are presented as mean + SD, **** = *p* value > 0.0001, and ns = not significant. Data were analyzed by one-way ANOVA with Tukey’s multiple comparison tests.

**Figure 3 ijms-25-00743-f003:**
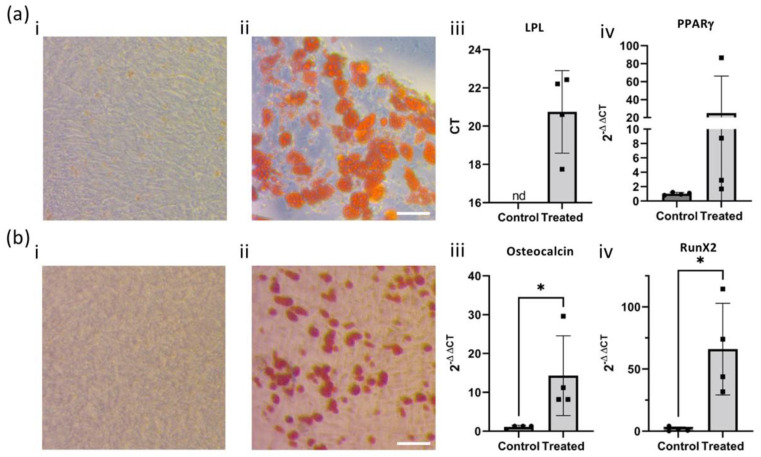
Unsorted hOM-MSCs can undergo adipogenesis and osteogenesis. Passage-matched, unsorted hOM-MSCs were treated with media sufficient to drive adipogenesis (**a**) or osteogenesis (**b**). (**a**) Oil Red O staining of lipid was performed on (**a**(**i**)) undifferentiated or (**a**(**ii**)) differentiated cells. Cells were also harvested for RNA, and RT-qPCR was performed with primers against key adipocyte-related genes (**a**(**iii**)) LPL and (**a**(**iv**)) PPARγ, with undifferentiated (control) and differentiated (treated) cells compared. (**b**) Alizarin Red S staining of calcium deposition was performed on (**b**(**i**)) undifferentiated or (**b**(**ii**)) differentiated cells, using the same microscope set-up as (**a**). Again, RNA extraction was performed on the control, and RT-qPCR was performed against relevant genes of interest (**b**(**iii**)) Osteocalcin and (**b**(**iv**)) RunX2. Images were captured by light microscopy at 400× magnification. RT-qPCR was analyzed (where possible) by the 2-DDCT method or otherwise expressed as raw CT, with values plotted as mean ± SD and statistical comparison between control and treated samples performed by an unpaired *t*-test (* = *p* value > 0.05). Each data point represents cell lines generated from an independent tissue donor. ‘nd’= not detected. Scale bar = 200 μm.

**Figure 4 ijms-25-00743-f004:**
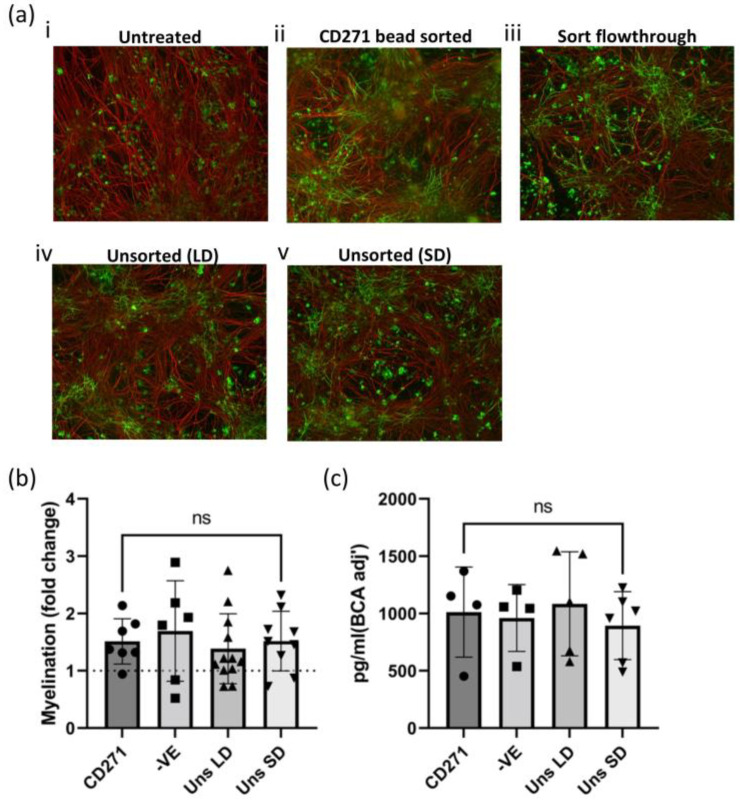
Unsorted hOM-MSCs are comparable to CD271-sorted cells in their myelinating potential. (**a**) Conditioned media (CM) was generated from hOM-MSCs; (**a**(**ii**)) CD271-sorted cells (CD271); (**a**(**iii**)) the negative fraction flowthrough from the sort (-VE); (**a**(**iv**)) unsorted cells seeded at a low starting density (Uns LD); and (**a**(**v**)) unsorted cells seeded at a standard starting density (Uns SD). These, along with unconditioned media (**a**(**i**)), were applied to myelinating cultures for a period of 14 days. After which, cultures were fixed and stained for axons (red) and myelin (green), blinded, and epifluorescent microscopy performed (100× magnification). (**b**) The percentage of axons myelinated was determined, normalized to cultures treated with non-conditioned media (represented as the dotted line at 1), and expressed as a fold change from these controls for each sort of condition. Each datapoint represents a single myelinating culture. (**c**) CXCL12 levels in this CM were determined by ELISA, normalized to whole protein lysate concentration from the cells used to generate it, and compared between sort conditions. Each datapoint represents CM generated from hOM-MSCs derived from an independent donor. Statistical comparisons were made by one-way ANOVA with Tukey’s multiple comparison tests. ns = not significant. Scale bar = 100 μm.

**Figure 5 ijms-25-00743-f005:**
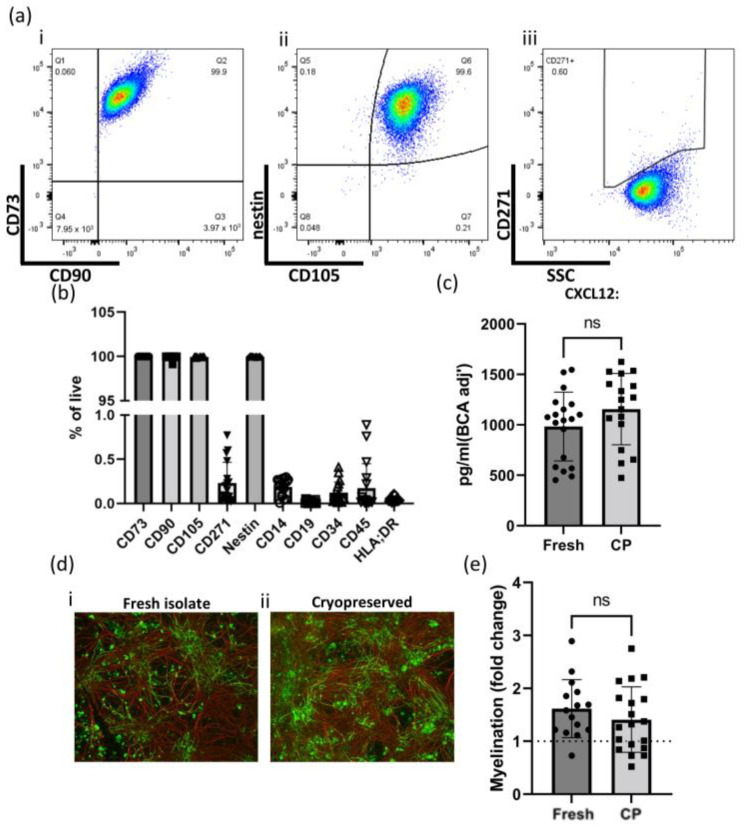
Cryopreservation has no notable effects on hOM-MSC surface marker expression, CXCL12 production, or myelinating potential. (**a**) hOM-MSCs were recovered from cryopreservation and evaluated by flow cytometry. Live singlets were analyzed for their expression of (**a**(**i**)) CD73/CD90, (**a**(**ii**)) nestin/CD105, and (**a**(**iii**)) CD271, with exemplar plots shown. (**b**) Expression of positive and negative MSC markers was determined for cell lines generated from each donor (6) and recovered from cryopreservation. (**c**) CM was generated from cryopreserved cells as before (CP), with levels of CXCL12 determined by ELISA and compared to those derived from freshly isolated cells (Fresh). Each datapoint represents CM generated from cell lines derived using the different methods described previously, from six independent donors. (**d**) CM from freshy isolated (**d**(**i**)) and cryopreserved and recovered (**d**(**ii**)) hOM-MSCs was used in myelinating cultures as before, fixed and stained for axons (red) and myelin (green), and imaged by epifluorescent microscopy (100× magnification). (**e**) The percentage of myelinated axons was determined and compared to that of freshly isolated cells, with values normalized to unconditioned media-treated controls (represented by a dotted line at 1) and expressed as fold change. Individual datapoints represent a single myelinating culture. (**b**,**c**,**e**). Values plotted as mean ± SD. (**c**,**e**). ns = not significant. Data analyzed by non-paired Student’s *t*-test. Scale bar = 100 μm.

**Figure 6 ijms-25-00743-f006:**
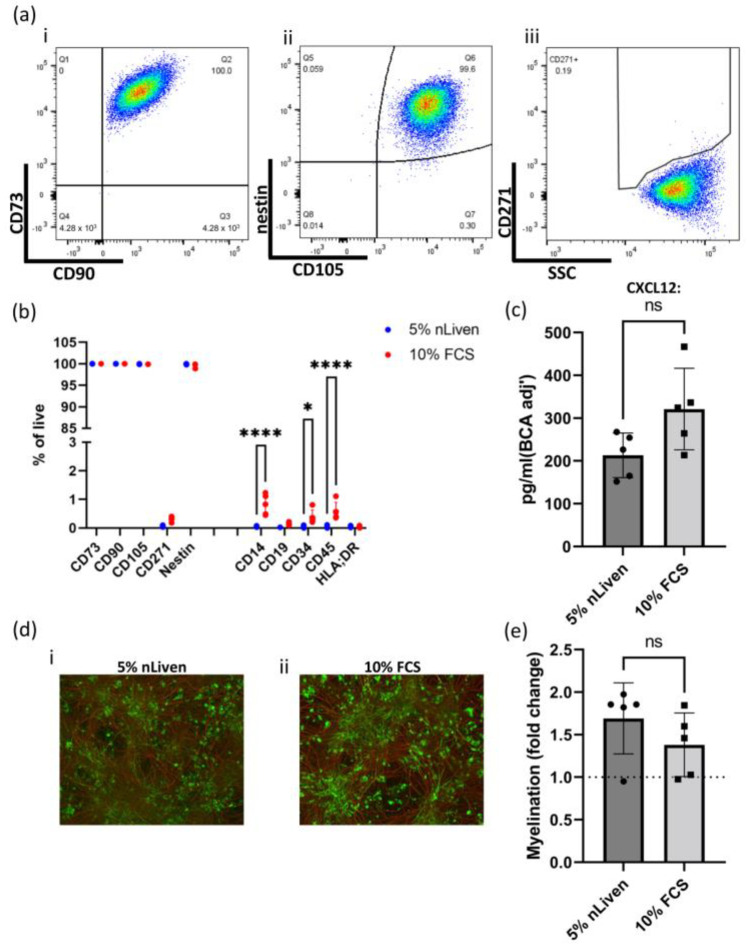
The choice of culture media formulation appears to have only minor effects on hOM-MSC surface marker expression and little to no impact on CXCL12 production or myelinating potential. (**a**) hOM-MSCs were grown in media supplemented with either 5% nLiven or 10% FCS and evaluated by flow cytometry. Live singlets were analyzed for their expression of (**a**(**i**)) CD73/CD90, (**a**(**ii**)) Nestin/CD105, and (**a**(**iii**)) CD271, with exemplar plots of cells grown in the presence of 10% FCS shown. (**b**) This was performed in parallel for 5 independent donor cell lines, with the average percentage of cells expressing positive (CD73, CD90, CD105, CD271, and nestin) and negative (CD14, CD19, CD34, CD45, and HLA-DR) MSC markers compared between cells expanded in either 5% nLiven or 10% FCS. (**c**) CM from each group was generated as before, and levels of CXCL12 were determined by ELISA and normalized by whole cell lysate concentration. (**d**) Myelinating cultures were performed using CM generated from cells expanded in (**d**(**i**)) 5% nLiven or (**d**(**ii**)) 10% FCS, with cultures fixed, stained for axons (red) and myelin (green), and imaged by epifluorescent microscopy (100× magnification). (**e**) The percentage of myelinated axons in these cultures was enumerated and normalized to unconditioned media controls (represented by a dotted line at 1), with comparisons made between the groups. (**b**,**c**,**e**) Values plotted as mean ± SD, with each individual datapoint representing a different original tissue donor. (**b**) Data analyzed by 2-way ANOVA with post-multiple unpaired *t*-tests. (**c**,**e**) Data analyzed by an unpaired Student’s *t*-test. **** = *p* value > 0.0001, * = *p* value > 0.05, and ns = not significant. Scale bar = 100 μm.

**Figure 7 ijms-25-00743-f007:**
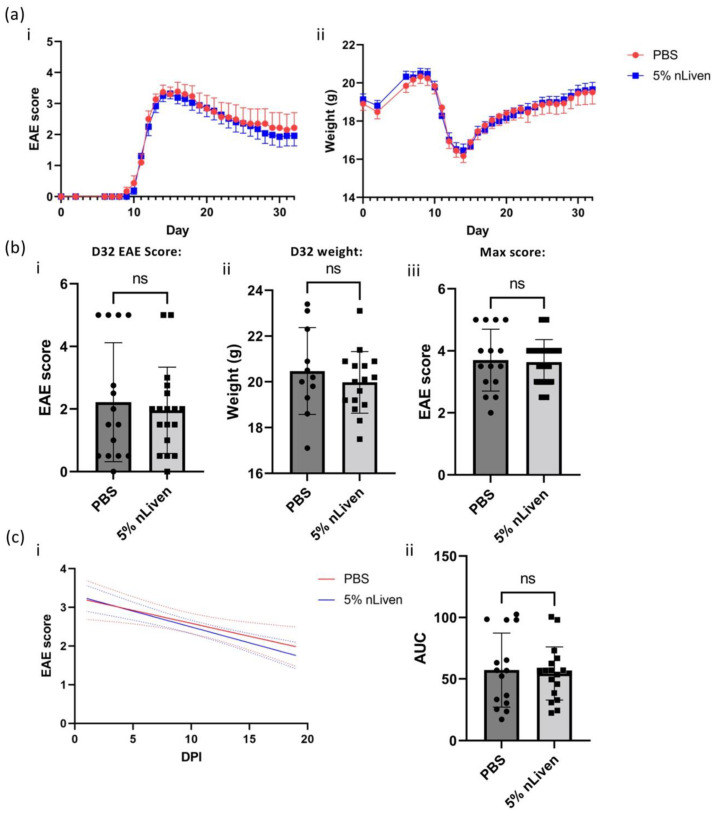
GMP-compatible hOM-MSCs do not improve disease outcomes in an in vivo model of demyelinating diseases. (**a**) EAE was induced in C57/BL6 mice, with animals injected intravenously with either PBS (n = 15, red line/symbols) or 1 × 10^6^ near GMP-compliant hOM-MSCs (5% nLiven, n = 18, blue line/symbols) soon after disease onset. Disease score (**a**(**i**)) and weight (**a**(**ii**)) were monitored throughout the disease course. (**b**) Comparisons were made between PBS and hOM-MSC-injected animals for their disease scores (**b**(**i**)) and weight (**b**(**ii**)) on day 32 (D32) of the clinical course, as well as the maximum clinical score observed for each animal during the model (**b**(**iii**)). (**c**) Further comparisons were drawn between control and treated animals by linear regression analysis of their disease scores (**c**(**i**)) from 1 day post-injection (DPI), as well as the area under the curve (AUC) of their disease scores over the full course (**c**(**ii**)). Combined analysis of two independent EAE experiments. In each experimental run, treated animals were injected with hOM-MSCs derived from five different donors (a minimum of one animal per donor). In all cases, each datapoint represents an individual animal, with data presented as mean ± SD and potential statistical significance determined by an unpaired Student’s *t*-test. ns = not significant. Dotted lines in (**c**(**i**)) represent SEM (PBS = red, 5%nLiven = blue).

**Table 1 ijms-25-00743-t001:** Antibodies used in the characterization of hOM-MSCs.

Marker:	Extra/Intracellular:	Clone:	Fluorophore:	Manufacturer:	mL/Sample:
CD73	Extracellular	AD2	PE	Biolegend	0.25
CD90	Extracellular	5E10	AF700	Biolegend	1
CD105	Extracellular	266	BUV395	BD Biosciences	1
CD271	Extracellular	ME20.4	PerCP/Cy5.5	Biolegend	1
Nestin	Intracellular	10C2	AF488	Biolegend	1
CD14	Extracellular	HCD14	PE/Cy7	Biolegend	0.5
CD19	Extracellular	HIB19	BV785	Biolegend	0.5
CD34	Extracellular	561	BV421	Biolegend	0.5
CD45	Extracellular	HI30	APC	Biolegend	0.5
HLA:DR	Extracellular	L243	BV510	Biolegend	0.5

**Table 2 ijms-25-00743-t002:** Primer sequences used in the characterization of hOM-MSCs.

GOI:	For/Rev:	Sequence:
GAPDH	F	GGTCACCAGGGCTGCTTTTA
GAPDH	R	TTCCCGTTCTCAGCCTTGAC
LPL	F	TCACAGCAGCAAAACCTTCATG
LPL	R	CAGCCAGTCCACCACAATGA
PPARg	F	AAGCCCTTCACTACTGTTGACTT
PPARg	R	GCAGGCTCCACTTTGATTGC
BGLAP	F	CCTCACACTCCTCGCCCTAT
BGLAP	R	CGCCTGGGTCTCTTCACTAC
RUNX2	F	AACAGCCTCTTCAGCACAGT
RUNX2	R	GCTCACGTCGCTCATTTTGC

## Data Availability

Raw data were generated at the University of Glasgow. Derived data supporting the findings of this study are available from the corresponding author [S.C.B.] upon reasonable request.

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
