# Peer review of "Development of Good Manufacturing Practice-Compatible Isolation and Culture Methods for Human Olfactory Mucosa-Derived Mesenchymal Stromal Cells"

_ijms, 2024, doi:10.3390/ijms25020743_

Round 1

Reviewer 1 Report

Comments and Suggestions for Authors

The selected EAE model has not been proved in the cited literature that hOM-MSCs can promote myelination, and the selection of the model here is not rigorous. Perhaps the author needs to reconsider the choice of model or explain the possible phenotypic changes of cells in depth, thus increasing the clinical significance of the article.

Author Response

We would like to thank the reviewer for taking the time to assess the work presented here. We have responded to their comments as follows (associated text changes in the manuscript are highlighted in pink):

We would like to thank the reviewer for highlighting this. Indeed, there isn’t direct evidence of hOM-MSCs enhancing remyelination in the EAE model and this should not have been inferred when referencing this previous work (Lindsay SL et al., 2022). The text has been amended to reflect this [line 79]. However, this previous study did demonstrate significantly improved disease outcomes when hOM-MSCs were administered at mild and more severe stages of EAE clinical progression. hOM-MSC treatment was also shown to reduce immune infiltration and axonal pathology, as well as increasing blood brain barrier integrity. As this work was conducted with cells established and grown in non-GMP compliant conditions, the decision was made to repeat this model using the cells obtained via the GMP-compatible methods we have since developed.

We would agree with the reviewer that other models would be useful in the assessment of GMP-compatible hOM-MSCs. We have previously transplanted non-GMP compliant hOM-MSC into a rat model of spinal cord injury and reported remyelination of spared fibres quicker than in controls (determined by immunolabelling of peripheral myelin protein 0 (P0) in spinal cord sections), correlated with improved gait recovery (Lindsay SL et al., 2017). However, to repeat this with GMP-compliant cells is out with the scope of this current work, which was principally intended as development and characterisation of GMP-compatible hOM-MSCs with only very preliminary in vivo characterisation being performed here. Such experiments certainly could not be performed in the 10-day review window.

There are only very mild phenotypic differences observed in the expression of negative MSC markers and a non-significant increase in CXCL12 production in cells grown in 10% FCS. We have amended the text to provide comment on these findings [line 429-437]. Likely, we would need to perform extensive additional experiments to assess the transcriptome/secretome of these cells to determine if there are any significant differences that could account for these phenotypic changes (which again, will not be possible within the review period given), and would also arguably be better served as the focus of a follow up study.

Reviewer 2 Report

Comments and Suggestions for Authors

It’s a well-written manuscript. I have following questions:

Please check the manuscript and correct small mistakes, for examples: P value > 0.05 in Figures 2, 3, 6. Also, what’s IV administration (Line 677). 

Methods: 

Please describe details of the following procedures: such as tissue size including weight (g?), and how many cells after minced? What’s the yield of MSCs at the P1 passage? Have you ever raised MSCs using your methods to clinical scales?

Interestingly, tissue can be minced into single cells, which generated cluster-like clones and developed into MSCs. Are these cell clusters floating or attach to the flask?

Please clarify whether antibiotics were used or not. Reading your abstract, the impression was no antibiotics used but in the method, P/S was added in MSC medium. 

Please clarify which size flask was used for MSC culture. If those MSCs are consider cell lines, how many maximum passages can you raise until they slow down? Are they continuing to be homogeneous, without morphological changes? 

Figure 4: which day or passage did you collect conditioned media (CM)? Have you done human cytokine/chemokine array (R&D) assay on these CM, besides CXCL12? If not, please either do it or discuss. 

Figure 7: although there is no significant therapeutic effect on EAE mice, did you investigate the histology of spinal cord or brain and changes of immune cells after MSC treatment? You need to provide some beneficiary evidence of either comparative histology or key cytokine release such as NGF, etc instead of releasing pro-inflammatory CXCL12 to treat autoimmune disease models, right? 

Please revisit your manuscript, try to avoid overstatement in introduction and last paragraph (Line423-429: Regardless….) before Methods. Myelin degeneration and regeneration are complex pathological conditions, affecting neural circuitry. Thus, overemphasizing CXCL12 is not appropriate. 

Author Response

We would like to thank the reviewer for taking the time to assess this work, and for their kind comments. We have checked the document again and corrected any mistakes where we have found them. These (and any other changes mentioned in the following) are highlighted in the reviewed manuscript in blue. In answer to their other queries:

  • Tissue size and weight were not assessed at time of processing. This did vary considerably as these were patients undergoing other surgical procedures from which opportunistic biopsies could be obtained, rather than biopsies of a distinct size taken for investigative purposes. For 5/6 samples, pre-processing images were taken (and can be fully anonymised and shared with the reviewer on request). Unfortunately, no scale bar was included, but all images were taken in 10 cm tissue culture plates, potentially allowing for a rough estimation of sample size. An additional point is that some of these samples did not arrive as a single piece, complicating the estimation of sample size. As for cell number, mechanical processing only minced tissue down to pieces of around 1-2 mm, and so a cell count would not be possible.

  • Total cell counts were obtained from each passage, including P1. The theoretical total yield was not calculated. However, this could be done using cell count data if this would be considered essential/beneficial to the manuscript.

  • Cells were not grown to clinical scale as we lacked access to the facilities (such as bioreactors) to do so, and for these experiments there was no requirement of cell numbers in excess of 107 at any given time. However, previously hOM-MSCs have demonstrated favourable growth kinetics (~8x that of hBM-MSCs, Lindsay SL et al., 2013), and so we would not predict any issue in growing hOM-MSCs to this scale.

  • As stated above, tissue was not minced into a single cellular suspension. An outgrowth approach was adopted here, in which non-adherent tissue and cells were removed midway through the initial outgrowth phase. Consequently, only adherent cells were captured.

  • Antibiotics were only utilised in the initial outgrowth phase for 3 samples, and in these instances the initial processed tissue was split and also grown in the absence of antibiotics. This was done in order to provide a control group for the endotoxin testing conducted on these samples. After this initial phase, and with no bacterial growth of detectable endotoxin found in any sample, donor matched outgrown cells were combined for subsequent passaging and analysis. We have amended the relevant material and methods section in a manner that clarifies this.

  • 10 cm tissue culture dishes were used in the initial outgrowth phase. 25 cm2 flasks were used for the first passage of CD271 sorted cells (due to lower cell numbers) and 75 cm2 flasks used in all other instances. The text has been amended to reflect this.

  • As this work was focused on the preparation of hOM-MSCs for potential future clinical applications, longevity in culture was not assessed (due to the increasing risk of aberrant cells emerging with extensive passaging). In all cases, cells were passaged up until P3, and where possible cryopreserved at each passage. However, we have published that huOM-MSC grow faster than bone marrow derived MSC (hBM-MSC) and can be both quickly and easily propagated in large enough numbers for transplantation (Lindsay et al., 2013).

  • CM was generated from P3 cells in all cases, the text has been amended to indicate this. A cytokine/chemokine array was not performed on these GMP-compatible cells. However, the secretome of hOM-MSCs has been previously assessed in comparison to hBM-MSCs using miRNA profiling and a Multiplex Chemokine array with 39 targets (Lindsay SL, et al., 2016). This analysis highlighted CXCL12 as a critical secretory factor in the pro-myelinating phenotype of hOM-MSCs. This finding was further validated by experiments demonstrating that treatment of myelinating cultures with CXCL12 in the media also increased myelination, and that pro-myelinating effects of CXCL12 could be abrogated in cultures treated with hOM-MSC CM or exogenous CXCL12 by the administration of pharmacological blockers and/or altering CXCL12 posttranscriptional control in hOM-MSCs during CM generation. Additionally, part of our aims with this manuscript is to establish potential release criteria for the use of these cells as a therapeutic, and the additional cost and complexity of performing such multiplex assays for each sample may not be suitable in this scenario. For these reasons, the decision was made to focus solely on CXCL12, and establish high production of this chemokine as part of the proposed release criteria. Consequently, we have amended the text both to better summarise these prior findings [line 57-69] and indicate our reasoning for selecting CXCL12 solely in our analysis of these cells [line 201-206]. We hope this satisfies this enquiry.

  • Unfortunately, tissue samples were not taken for histology, and the immune cell content of these tissues was not assessed. In one run of the model, peripheral immune cells were harvested from spleen and inguinal lymph nodes but were cryopreserved for potential future experiments and not evaluated at this time. However, given these were harvested at the end of the model where clinical score was improving, and that scores were comparable between the treated and control groups, we believe it is unlikely that potential differences in peripheral immune profile would be observable. On the matter of NGF release by hOM-MSCs, this has not been assessed here. In the previously discussed study in which the chemokine/cytokine multiplex assay was utilised, NGF was detected in the secretome as well using a Human Multi-Neurotrophin Rapid Screening ELISA Kit (Lindsay SL et al., 2016). However, myelination was reduced to controls levels following neutralisation of CXCL12 alone here, suggesting NGF was not necessary for the promotion of myelination in these assays.

While we could potentially assess CM from hOM-MSCS for NGF, it isn’t possible to do so within the review period and would only confirm that this was secreted by these cells in culture rather than produced in vivo.

Ultimately, the purpose of this study was to develop GMP-compatible methods for the isolation and expansion of hOM-MSCs, characterise them in vitro and conduct an initial in vivo assessment. While we have demonstrated that GMP-compatible hOM-MSCs meet the criteria for classification as MSCs and retain previously established characteristics of hOM-MSCs in vitro, we must conclude from these experiments that (unlike what has been previously observed with non GMP-compliant cells, Lindsay SL et al., 2022) GMP-compliant hOM-MSCs do not demonstrate in vivo efficacy in EAE. We do agree with the reviewer that another means of establishing beneficiary evidence of these cells is needed. However, it would likely require the use of a different model in which an observable clinical benefit of hOM-MSCs was present and we believe this lies out with the scope of this current study.  

  • We have amended the introduction [line 35-38] and concluding paragraph [line 450] to better convey that any future use of hOM-MSCs is likely to be part of a combinatorial approach to the treatment of CNS pathologies, rather than being proposed as a singular treatment in demyelinating diseases. While we have highlighted CXCL12 here, as mentioned in a previous response we believe our prior findings substantiates this as being critical specifically in the pro-myelinating phenotype seen with hOM-MSCs in in vitro cultures, and believe our reasons for proposing it as a potential release criteria for hOM-MSCS (that it is a notable difference between these cells and others and is critical to their in vitro pro-myelinating phenotype and easily measurable) are justified. We appreciate the reviewers concern of potential overreach; but we do not believe our summary paragraph is an overstatement. We have demonstrated that hOM-MSCs meet ISCT criteria can be developed in a GMP-compatible manner, and do not mention CXCL12 here or claim any efficacy of GMP-compliant hOM-MSCs in demyelinating disease in this study. We certainly did not intend to imply CXCL12 was critical in tackling complex CNS pathologies and hope our previously discussed reasoning and text changes regarding it’s inclusion here ameliorates this concern.

Reviewer 3 Report

Comments and Suggestions for Authors

The Article is devoted to the analysis of human olfactory mucosa mesenchymal stromal cells (hOM-MSCs) obtained using a good manufacturing practice (GMP)-compatible approach, which is very important to study before the cells would be considered for clinical use. The authors demonstrated the low effect of cryopreservation on the phenotype of hOM-MSCs grown in GMP conditions which is important to know in case of potential utilization of the cells. The efficacy of hOM-MSCs-derived conditioned media or cells themselves was evaluated in vitro and in vivo respectively.

The research is planned very logically and described consistently; each detail was taken into account. The obtained results are analyzed profoundly, it has been demonstrated that GMP-compliant hOM-MSCs growth, expression of typical MSC surface markers, and secretion of CXCL12 are satisfactory and further preclinical evaluation is offered to study the effect of hOM-MSCs on disease progress.

The following comments do not diminish the value of the Article:

Line 32 Probably it would be better to decipher the abbreviation CNS (central nervous system) at the first mention in the main text also.

Line 68 Probably it would be also good to decipher the abbreviation of the EAE - an in vivo model of demyelinating disease (experimental autoimmune encephalitis) – in the same form as it’s been given in the Abstract.

Line 109 Probably it would be better to decipher the following abbreviation: ISCT.

Line 141 Would you please describe the following abbreviation: -VE.

Line 143 Does the following symbol ‘****’ demonstrate the comparison with Passage 0 (d, iii)?

Line 198 Please check the titles of comparable sort conditions (Figure 4 b, c).

Please check the description of the references as it should be prepared according to the Journal's requirements.

Author Response

We greatly appreciate the reviewer’s very kind comments on the article. In response, the following amendments have been made (highlighted in yellow):

  • All explanations for the mentioned abbreviations have been added to the text
  • Group names in figure 4 b&c have been amended to match those used previously in figure 1
  • Apologies, we selected the default numbering style in Endnote, rather than the IJMS specific one. This has now been amended, however was not highlighted like the other changes as it would have been the entire reference list